# Improved visualization of high-dimensional data using the distance-of-distance transformation

**Jinke Liu** [1,2]*, **Martin Vinck** [1,2]

**1** Ernst Strüngmann Institute for Neuroscience in Cooperation with Max Planck Society, Frankfurt am Main, Germany, **2** Donders Institute for Brain, Cognition and Behaviour, Nijmegen University, Nijmegen, Netherlands

* jinke.liu@esi-frankfurt.de

## Abstract

Dimensionality reduction tools like t-SNE and UMAP are widely used for high-dimensional data analysis. For instance, these tools are applied in biology to describe spiking patterns of neuronal populations or the genetic profiles of different cell types. Here, we show that when data include noise points that are randomly scattered within a high-dimensional space, a "scattering noise problem" occurs in the low-dimensional embedding where noise points overlap with the cluster points. We show that a simple transformation of the original distance matrix by computing a distance between neighbor distances alleviates this problem and identifies the noise points as a separate cluster. We apply this technique to high-dimensional neuronal spike sequences, as well as the representations of natural images by convolutional neural network units, and find an improvement in the constructed low-dimensional embedding. Thus, we present an improved dimensionality reduction technique for high-dimensional data containing noise points.

## Author summary

Biological datasets are often high-dimensional, e.g. the genetic profile of cells or the firing pattern of neural populations. Dimensionality reduction methods like t-SNE are commonly used to represent the high-dimensional data in a low-dimensional embedding space. The visualization helps us to identify the underlying clustering patterns and shed light on the information hidden within the data. We show that in situations where there exist scattering noise points, clustering patterns in the data tend to be heavily distorted. Here, we show that using a distance-of-distance (DoD) transformation of the dissimilarity matrix between data points, the influence of scattering noise is effectively removed. This neighborhood-based transformation is most effective when the dimensionality of the dataset is high. We show that this technique improves low-dimensional embedding for several high-dimensional datasets, such as the convolutional neural network representation of natural images or the neuronal population representation of visual stimuli.

**Data Availability Statement:** The paper uses publicly electrophysiological data from Allen Institute Brain Observatory 1.1 (https://allensdk.readthedocs.io/en/latest/visual_coding_neuropixels.html). The images are taken from

public available ImagNet dataset (https://www.image-net.org/) and Sketch dataset (https://sketchy.eye.gatech.edu/). All code is shared at a github repository at https://github.com/Jinke-Liu/Distance-of-Distance-tSNE.

**Funding:** This project was supported by a BMBF Grant to M.V. (Computational Life Sciences, project BINDA, 031L0167). The funders had no role in study design, data collection and analysis, decision to publish, or preparation of the manuscript.

**Competing interests:** The authors declare that no competing interests exist.

This is a *PLOS Computational Biology* Methods paper.

## Introduction

A major goal of data science is to extract patterns from high-dimensional data containing multiple features. It is typically required to construct a low-dimensional representation of high-dimensional data for the purpose of visualization, noise reduction, or feature extraction. In fields such as biology, where high-dimensional data sets are common, dimensionality reduction approaches are widely adopted. For instance, in neuroscience, dimensionality reduction techniques have been used to study the way in which neuronal populations represent motor and visual information [1–3]. It is also a standard approach to study the the genetic profiles of different cell types [4–7]. Dimensionality reduction techniques based on embeddings including t-SNE [8, 9] and UMAP [10] have been developed to represent high-dimensional data with only two or three components. The principle underlying these techniques is to treat data points as particles that are attracted to their neighbors and repelled by distant data points. Despite their usefulness, it is known that algorithms like t-SNE have inherent limitations, such as: sensitivity to hyper-parameters like perplexity; difficulty to capture global structure in the data especially when there are many clusters [11]. Therefore, it is important to optimize the pre-processing of the data and application of low-dimensional embedding techniques [7].

Here we show another problem with methods like t-SNE, namely that its performance strongly deteriorates when the data set contains many noise points. We show that the low-dimensional embedding space can become crowded due to the presence of noise points. The basic mechanism is that noise points repel each other and therefore start overlapping with clusters, even though the noise points have large distances to the clusters. As a result, meaningful patterns in the data can be masked. To our knowledge, there exists no simple solution to this "scattering noise problem". Although clustering techniques like HDBSCAN can be used to identify noise points [12], these techniques do not solve the scattering noise problem in terms of low-dimensional visualization. Furthermore, although in some situations PCA may aid to denoise the data, PCA can also remove important information from high-dimensional data sets. As we will show, PCA does not in general effectively solve the scattering noise problem.

Here, we present a simple technique to solve the scattering noise problem. We show that scattering noise problem for high-dimensional datasets can be effectively alleviated with a transformation of the distance matrix. We call this the distance-of-distance (DoD) transformation, because it considers the differences between distances in a certain neighborhood of the data points. We apply the DoD transformation to both electrophysiological recordings of neurons in the mouse visual cortex during the presentation of drifting grating stimuli, as well as representations of natural image patches by convolutional neural network units. We demonstrate that in both cases, the DoD transformation facilitates the separation between noise points and cluster points in the low-dimensional embedding space.

## Materials and methods

### Simulation

We generated high-dimensional cluster points and noise points by randomly sampling from the multivariate Gaussian distributions with a standard deviation of 0.1. We drew points within one cluster from the same Gaussian distribution, while noise points were independently distributed. Then, we calculated the distance matrix between all pairs of data points by using

either euclidean or city-block metric. To find the low-dimensional embeddings, we applied t-SNE algorithm with perplexity value ranging from 5 to 50 and initialization with different random seed. We used an open implementation of t-SNE algorithm from sklearn (version 0.23.2). In some settings, we adopted PCA initialization combined with PCA preprocessing (S3 Fig).

To analyze how dimensionality and number of points influenced the performance of DoD transformation, we simulated cluster points from Gaussian distributions and noise points randomly distributed in a hyper-cube. Next, to compute the cluster-to-cluster distance, we considered all pairs of points, each from a different cluster, then we calculated the average distance of all such pairs. To compute the cluster-to-noise distance, we considered all pairs of points, such that one is a noise point and the other from a cluster, and then calculated the average distance between all such pairs. To compute the noise-to-noise distance, we considered all pairs of noise points, then we calculated the average distance of all such pairs. Next, we used DoD transformation to manipulate the original pairwise distance matrix $\mathcal{D}$. After the manipulation, we obtained the distance-of-distance matrix $\mathcal{F}$. We measured the distance shrinkage of all three types of distances (cluster-to-cluster, cluster-to-noise, and noise-to-noise) in two ways: Either the absolute shrinkage was calculated as $\Delta = \bar{d} - \bar{f}$, or the fraction was calculated as $\bar{f}/\bar{d}$. Therefore, a larger delta distance or a smaller fraction indicates a larger shrinkage. In order to measure the clustering performance, we adopted the commonly used metric Adjusted Rand Index (ARI).

## K nearest neighbor classifier

We used a K nearest neighbor classifier to measure the performance of the DoD transformation on noise-free datasets. We chose the optimal parameter $K$ of KNN based on cross-validated classification score. Then with the optimal parameter, we built the KNN model on both the original distance matrix and the distance matrix after the DoD transformation. We then compared the 5-fold cross-validated score of the classifiers.

## Neural data analysis

We analyzed neural data from area V1 obtained via electrophysiological Neuropixel recordings (Allen Institute, [13]). The drifting grating visual stimulus consists of a full-field sinusoidal grating that moves in a direction perpendicular to the orientation of the grating. The spatial-temporal frequency of the drifting grating stimulus is not considered in our study. In the public dataset provided by Allen Institute, the drifting grating stimulus moves in 8 different directions. They were shown to the animal in a random order (S7(A) Fig). Example raster plots were taken from drifting grating response of session 754829445. We selected the visual neurons with high signal to noise ratio (snr $\geq$ 0.3), and the total number of neurons were 191. We applied SPOTDisClust algorithm [14] on the population spiking patterns within 100 ms after the stimulus onset. We used the output SPOTDis matrix for the following t-SNE analysis. Please refer to S7 Fig to find more details.

## Image data analysis

Images were obtained from the ImageNet data set. We used the pretrained VGG16 [15] as the convolutional neural network model. We cropped original images to create image patches that match the input size expected by VGG16. For each image patch, we extracted the responses of artificial neurons in the fully connected layer (fc6) as its representation. The dimensionality of the feature vector is 4096. Code is available at Github.

## Results

### Simulation

There are various techniques to construct a low-dimensional embedding of high-dimensional data, such as t-SNE [8, 9] and UMAP [10]. These unsupervised techniques are commonly used to visualize the results of clustering and to study the geometry of high-dimensional data. For some applications, however, a part of the data set could be comprised of noise points that are randomly scattered in a high-dimensional space. For example, the activity pattern of a high-dimensional neuron ensemble might show consistent clustering when the neural response is driven by specific stimuli, but could otherwise exhibit random behavior during spontaneous activity. When a dataset contains many noise points, the t-SNE and UMAP embedding exhibit a typical "scattering noise problem" (Fig 1C). That is, the noise points tend to be spread uniformly in the low-dimensional embedding space and are located near the clusters, despite the fact that the noise points are well separated from the cluster points. This scattering noise problem occurs because the noise points have, on average, a large distance between themselves, which causes them to repel each other. Thus, noise points can end up near or in a cluster region and can effectively mask the clusters that are present in the data set. Here, we develop a technique to address the scattering noise problem using a transformation of the distance matrix. We will show that the performance of low-dimensional embedding techniques is often improved by such transformation of the distance matrix. We start from a scenario where there are clusters, but also noise points that are scattering in a high-dimensional space. Consider the distance of a noise point $P$ to its nearest neighbors. In a high-dimensional space, we expect another noise point $Q$ to have a similar distance to the nearest neighbors of $P$ as $P$ itself. In other words, even though the distance between two noise points $P$ and $Q$ can be large, their distances to their respective nearest neighbors might in fact be very similar in a high-dimensional space. This is simply due to the fact that scattering points do not have particular clustering patterns, which makes any point almost identical to others in terms of their neighboring distance structure. This observation led us to compute the differences between the distances (i.e. the distance-of-distance), such that the scattering nature of noise points can be better captured after the transformation. Specifically, for each pair of data points, we consider the joint set of $K$ neighbors of these two data points, and then compute the distance-of-distance w.r.t. this set of neighbor points.

Mathematically speaking, given a dataset $X \in \mathbb{R}^{N \times D}$ with $N$ samples and $D$ features, the distance matrix is constructed as $\mathcal{D} \in \mathbb{R}^{N \times N}$ with either L1 or L2 distance metric, where $d_{i,j} = \|X_i - X_j\|_1$ or $\|X_i - X_j\|_2$. For any two data points $X_i, X_j \in \mathbb{R}^D$, we found their sets of $K$ nearest neighbors $I$ and $J$ respectively. Then we manipulated the original distance matrix $\mathcal{D}$ to yield another new distance matrix $\mathcal{F}$. We take the average absolute difference between the two points' distances to the selected neighborhood as the new distance as following,

$$f_{i,j} \equiv \frac{1}{2K} \left( \sum_{n \in I} |d_{n,i} - d_{n,j}| + \sum_{m \in J} |d_{m,i} - d_{m,j}| \right) \tag{1}$$

Subsequently, the distance-of-distance matrix $\mathcal{F}$ is used to substitute the original distance matrix used in the t-SNE algorithm. Applying this transformation, we should be able to obtain a high similarity between noise points, while retaining the distances from noise points to the clusters, and the distances between the clusters. Moreover, we expect the method to be relatively insensitive to the choice of neighborhood size $K$. As long as $K$ is kept smaller than the cluster size, the average density of the cluster neighborhood should not change much, therefore, the shrinkage of noise-cluster and noise-noise distances should in general apply (S1 Fig).

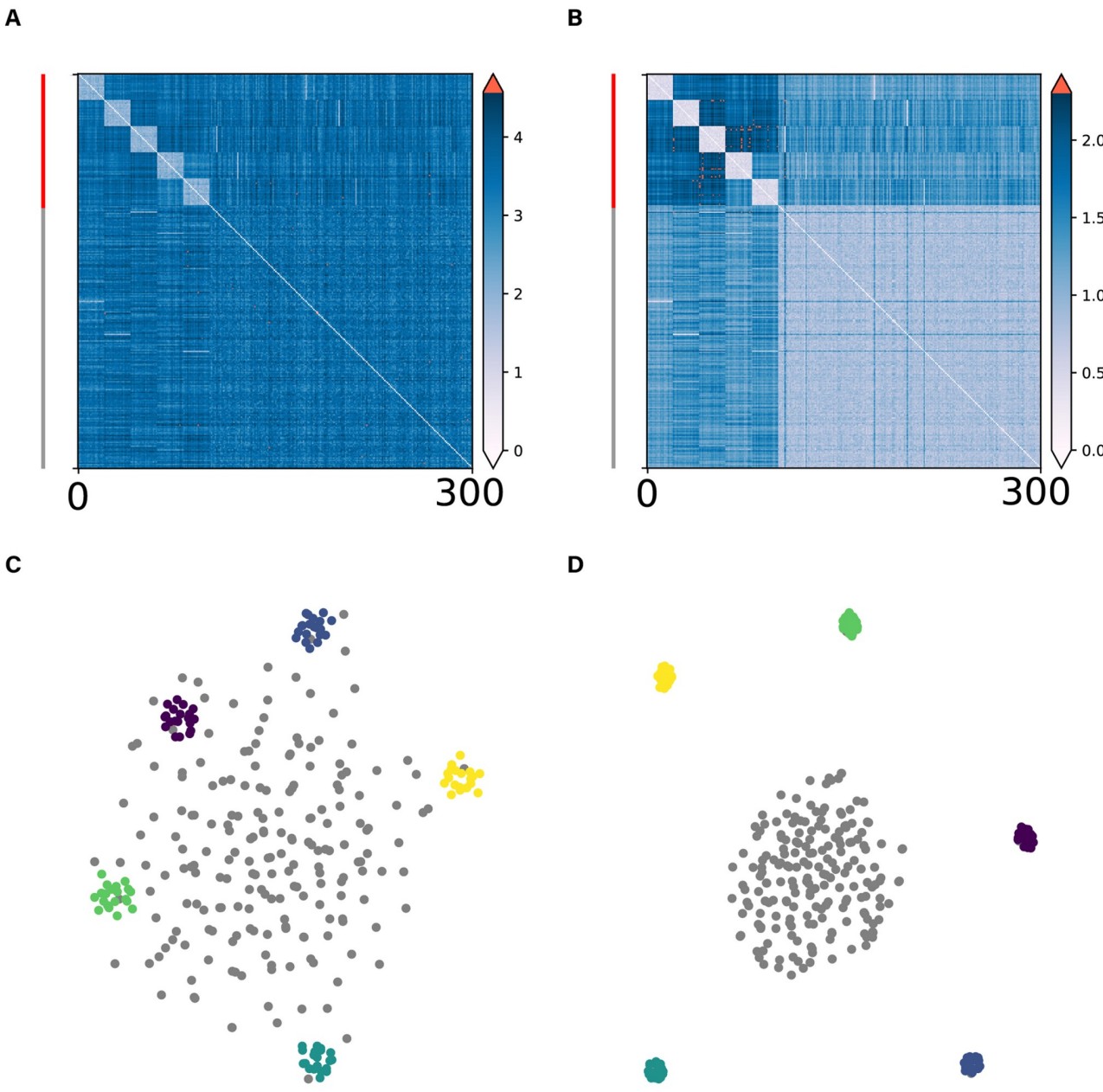

**Fig 1. Scattering noise problem in t-SNE and its alleviation through the DoD transformation.** In this simulation, we generated data points for five clusters and one noise cluster. For each of the five clusters, we sampled 20 points from different multivariate Gaussian distributions in a high-dimensional space ($D = 50$). Another 200 noise points were sampled uniformly from the same space. **A**: The original Euclidean (L2) distance matrix between the data points. Data points were ordered by clusters. **B**: The dissimilarity matrix after the DoD transformation. We computed the distance-of-distances for all pairs of points w.r.t. their nearest 10 neighbors. Red bar on the left indicates cluster points and black bar indicates scattering noise points. **C**: t-SNE visualization based on the original dissimilarity matrix. **D**: t-SNE visualization based on the dissimilarity matrix after the DoD transformation. Points from different clusters are labelled in different colors, noise points are labelled in grey.

We illustrate this behavior as an example (Fig 1C), where in the t-SNE embedding, noise points are located near or in the cluster regions. After the DoD transformation, the noise points attract each other, and do not randomly scatter over the low-dimensional embedding anymore (Fig 1D). Consequently, compared with the standard t-SNE, the noise points form a separate cluster that is isolated from the cluster points, thereby providing a better match with

cluster labels. Even in simulations where there are fewer noise points than cluster points in the data set, the DoD transformation was still robust (S4 Fig). Furthermore, the DoD transformation helps to separate the noise cloud from the clusters regardless of which distance metric we use to construct the original distance matrix $\mathcal{D}$. The results obtained by using L2 metric are almost identical to those by using L1 metric. Moreover, in situations where the noise labels are unknown, we can infer their identity based on the DoD transformation. By comparing the magnitude of the distance changes, it provides us with an automated way of denoising data (S2 Fig).

Intuitively, we expect the effectiveness of the DoD transformation to be dependent on two factors, namely the dimensionality of the data set and the number of data points. When the number of noise data points is relatively large compared to the dimensionality of the data, we expect that the DoD transformation has a minor effect, because two noise points will show relatively dissimilar distances to their respective neighbors. For example, on a one-dimensional line, if we take two out of many random noise points, then each noise point will have a nearby neighbor, and the distance-of-distances will be similar to the original distance. However, when the dimensionality of the data is relatively high compared to the number of noise points, we expect that the DoD transformation makes a large difference compared to the original distance. To investigate this, we performed several simulations with a varying number of cluster/ noise points and dimensionality. In these simulations, we drew noise points and cluster centers from a Gaussian distribution in a $D$-dimensional space (with a diagonal covariance matrix). Cluster points were generated from Gaussian distributions around the cluster centers. We observed that when the dimensionality of the feature space was low, the DoD transformation had minor effects on the low-dimensional embedding and on the distance matrix. However, when the dimensionality of the feature space was relatively high, the DoD transformation was highly effective in separating the cluster points from the noise points (Fig 2).

There are other techniques, such as principal component analysis (PCA), that are commonly used in combination with t-SNE. We showed that the scattering noise problem cannot be solved by simply using PCA for either initialization or preprocessing (S3 Fig). Furthermore, perplexity is a parameter in t-SNE algorithm that controls how near a point needs to be in order to be considered as a neighbor to a given point. We showed that by simply tuning perplexity, the scattering noise problem cannot be solved (S5 Fig).

Another degree of freedom in the DoD transformation comes from the parameter $K$, which controls the neighborhood size. We showed that our method is generally robust to the choice of $K$, even in situations where it is slightly larger than the cluster size. But unsurprisingly, the method fails when $K$ approaches the total number of points in the data set (S1 Fig). In situations where there are no scattering noise, applying DoD transformation to only clustering data introduced very limited distortion (S6(A) Fig). Even in situations where one cluster has lower density than the others, the DoD transformation did not significantly change the geometry of the low-dimensional manifolds (S6(B) Fig).

## Theoretical analysis

The simulations shown in Fig 2 suggest that the DoD transformation is most effective when the dimensionality of the feature space is relatively high. In this section, we will formalize the notion of the DoD transformation and explain why it improves the performance of low-dimensional embedding techniques such as t-SNE.

Suppose that a data set consists of $N$ points in a $D$-dimensional feature space. Each data point can be represented by a $D$-dimensional vector. Furthermore, suppose that the noise points are uniformly scattered in a $D$-dimensional hyper-cube and that there are several

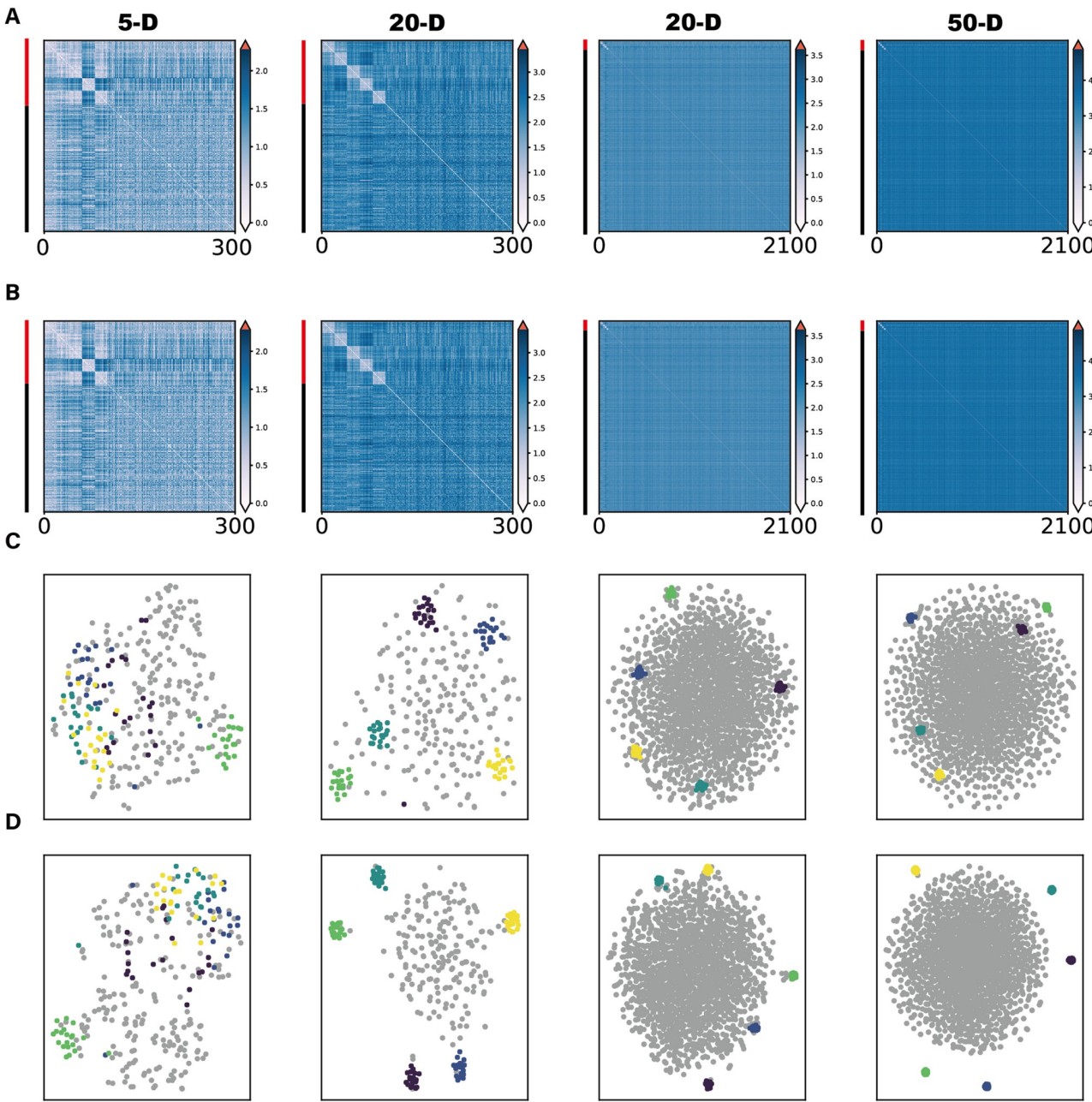

**Fig 2. Influence of dimensionality and number of noise points on the performance of the DoD transformation.** In these simulations, 20 cluster points were sampled from five different multivariate Gaussian distributions. The dimensionality was, from left to right, 5, 20, 20, and 50. The number of noise points was, from left to right, 200, 200, 2000 and 2000. distance-of-distances were computed w.r.t. 5 neighbor points. **A**: The original dissimilarity matrix. **B**: The dissimilarity matrix after the DoD transformation. Red bar on the left indicates cluster points and black bar indicates scattering noise points. **C**: t-SNE visualization based on original dissimilarity matrix. **D**: t-SNE visualization based on the dissimilarity matrix after the DoD transformation. Points from different clusters are labelled in different colors and noise points are labelled in grey.

clusters whose cluster centers are also uniformly scattered. To simplify our analysis, we will assume that the clusters have infinite density. In other words, points that belong to the same cluster have a mutual distance of zero. We will use the normalized L1 norm (i.e. the Manhattan distance metric divided by the dimensionality) to measure the distances between any pair of

data points. Since the distance is normalized by dimensionality, the distance will be finite for a finite volume, as $D \to \infty$. The analysis can be generalized to the L2 norm (Euclidean distance metric), but we use the L1 to make the analytical derivation easier. Consider that we have $M$ clusters $\mathbb{C}_m$ and a set of noise points $\Sigma$. Consider a pair of cluster and noise points $j \in \mathbb{C}_m$ and $\sigma \in \Sigma$ with a distance of $d_{j,\sigma}$. Let $j^*$ be the nearest neighbor such that $d_{j,j^*}$ is the L1 distance between point $j$ and its the first nearest neighbor. We consider a simple case where the DoD transformation has a joint neighborhood size of 2. Applying the DoD transformation with neighborhood size of 2, the distance-of-distance $f_{j,\sigma}$ between a noise and a cluster point can be expressed as

$$f_{j,\sigma} = \frac{1}{2}|d_{j,j^*} - d_{\sigma,j^*}| + \frac{1}{2}|d_{j,\sigma^*} - d_{\sigma,\sigma^*}|. \tag{2}$$

Because we assumed that the cluster is infinitely dense, the equalities $d_{j,j^*} = 0$ and $d_{\sigma,j^*} = d_{\sigma,j}$ hold. Hence,

$$f_{j,\sigma} = \frac{1}{2}d_{\sigma,j} + \frac{1}{2}|d_{j,\sigma^*} - d_{\sigma,\sigma^*}|. \tag{3}$$

If there are in total $N$ points scattering uniformly in a unit volume, the average distance of a point to its first nearest neighbor can be approximated [16] as

$$\mathbb{E}\{d_{\sigma,\sigma^*}\} \approx \frac{1}{3}\left(\frac{1}{N}\right)^{\frac{1}{D}}, \tag{4}$$

where the factor $\frac{1}{3}$ takes into account that we compute the normalized L1-distance. Note that as $D \to \infty$, $d_{\sigma,\sigma^*} \to \frac{1}{3}$, the expected (normalized) L1-distance between any two points that are uniformly distributed in a hyper-cube.

Because $\sigma^*$ is another random noise point, we have $\mathbb{E}\{d_{j,\sigma^*}\} \approx d_{j,\sigma}$. Furthermore, $d_{j,\sigma}$ will be larger than $d_{\sigma,\sigma^*}$, because $\sigma^*$ is the first neighbor of $\sigma$ (assuming that the first neighbor of the noise point is not a cluster point). Therefore, we can simplify the expression of the distance-of-distances $f_{j,\sigma}$ as

$$\mathbb{E}\{f_{j,\sigma}\} \approx \frac{1}{2}\left(2d_{j,\sigma} - \frac{1}{3}\left(\frac{1}{N}\right)^{\frac{1}{D}}\right). \tag{5}$$

Now consider two noise points $\sigma \in \Sigma$, $\epsilon \in \Sigma$ together with their nearest neighbors $\epsilon^*$, $\sigma^*$, and apply the same argument there. We have

$$f_{\sigma,\epsilon} \approx \frac{1}{2}|d_{\sigma,\epsilon^*} - d_{\epsilon,\epsilon^*}| + \frac{1}{2}|d_{\epsilon,\sigma^*} - d_{\sigma,\sigma^*}|. \tag{6}$$

Note that $\mathbb{E}\{d_{\sigma,\epsilon^*}\} \approx d_{\sigma,\epsilon}$ and that $\mathbb{E}\{d_{\epsilon,\sigma^*}\} \approx d_{\epsilon,\sigma}$. Hence the transformed distance can be expressed as

$$\mathbb{E}\{f_{\sigma,\epsilon}\} \approx \frac{1}{2}\left(2d_{\sigma,\epsilon} - \frac{2}{3}\left(\frac{1}{N}\right)^{\frac{1}{D}}\right). \tag{7}$$

Finally, we can see that the transformed distance between two points from two different clusters or from the same clusters will be identical to the original distance, given the assumption that a cluster is infinitely dense.

Thus, compared to the original distance, the distance-of-distance between two noise points decreases more strongly than the distance between a cluster and a noise point, by the amount

of 1/6 $(1/N)^{1/D}$, while the distance between two cluster points is preserved. It can be further seen that when $D \to \infty$, $(1/N)^{1/D} \to 1$ and therefore $f_{\sigma,\epsilon} \to 0$. Hence, as $D$ approaches infinity, we obtain the asymptotes $\mathbb{E}\{f_{\sigma,\epsilon}\} \to 0$ and $\mathbb{E}\{f_{j,\sigma}\} \to 1/6$.

Therefore, in very high-dimensional spaces, the DoD transformation preserves the geometrical distances between the cluster points, but pushes the noise points together, while relatively preserving distance between the clusters and the noise points. For a low-dimensional embedding technique, this means that the noise points will now be attracted to each other and be repelled by the clusters. Suppose that we want the difference in distance shrinkage between cluster-to-noise and noise-to-noise point to be greater than a threshold $\theta$. The resulting inequality shows a linear dependence on $D$ but a logarithmic dependence on $N$:

$$\frac{1}{6}\left(\frac{1}{N}\right)^{\frac{1}{D}} > \theta \Leftrightarrow \log N < -6D \log \theta . \tag{8}$$

## Dimensionality and number of points influence distance-of-distances

We performed further simulations to support these theoretical analyses. In the first simulation, we examined how the DoD transformation affects the distance between noise and cluster points. The theoretical analysis above predicts that if the dimensionality $D$ grows, the distance between pairs of noise points should show a relatively strong decrease, whereas the distance between a cluster and a noise point should show a relatively small decrease as compared to the original distance. To test this, we generated data using Gaussian mixture models. We then examined how the dimensionality and the number of noise points affects the distance-of-distances between cluster and noise points, two noise points and two points belonging to different clusters. As predicted, due to the DoD transformation, the distances between data points shrink. Furthermore, the average shrinkage of the distance between two noise points was larger than the case for a cluster point and a noise point. In addition, the shrinkage of distances increased as a function of dimensionality $D$, and decreased as a function of number of points $N$. (Fig 3).

## DoD transformation improves clustering

Next, we examined whether the DoD transformation improves clustering performance on the t-SNE embedding. To study this, we generated high-dimensional data from Gaussian distributions with different number of data points and different dimensionality. We then created low-dimensional embeddings and used the K-means algorithm to identify clusters in the t-SNE embeddings. To measure the clustering performance, we compared the true cluster labels with the inferred cluster labels using the Adjusted Rand Index (ARI). Fig 4A and 4B show how the DoD transformation improves the clustering performance and the t-SNE embedding. We found that the DoD transformation strongly alleviated the scattering noise problem in the low-dimensional embedding and improved the clustering performance, as measured by ARI. As the dimensionality $D$ of the data increased, the ARI score strongly increased. Conversely, the ARI score decreased as a function of the number of noise points (Fig 4C). As predicted from our theoretical analyses, clustering performance showed an approximately linear dependence on $D$ and a logarithmic dependence on $N$. This was indicated by the presence of a diagonal line in the heat map of ARI changes. This analysis shows that the DoD transformation improves the performance of clustering with the existence of scattering noise points, especially for large $D$ and a small number of data points $N$.

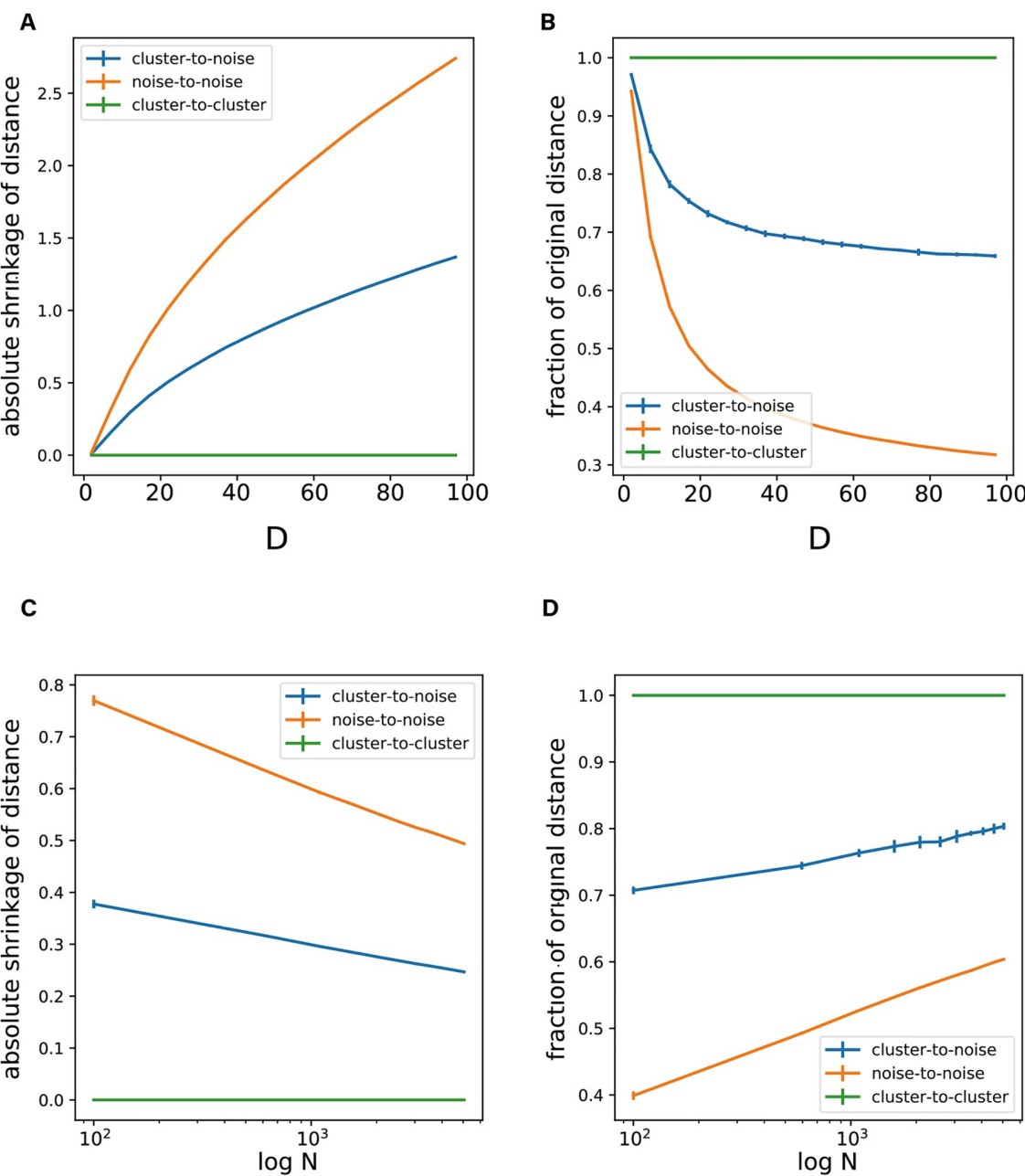

**Fig 3. Influence of dimensionality and number of points on distance-of-distances. A-B**: Influence of dimensionality $D$ on distance-of-distances. **A**: Absolute shrinkage of Euclidean distance (i.e. original distance minus the distance-of-distances) as a function of dimensionality $D$. **B**: Fraction of original distance (distance-of-distances divided by original distance) as a function of dimensionality $D$. The distance between cluster points is unaffected by the Distance-to-Distance transformation (because the clusters had infinite density). Because of the DoD transformation, the distance between noise points shrinks more than the distance between cluster and noise points. As a result, noise points become relatively more similar to each other than to other cluster points. **C-D**. Influence of the number of points $N$ on distance-of-distances. **C**: Absolute shrinkage of distance as a function of number of points $N$. **D**: Fraction of original distance as a function of number of points $N$. As the number of data points increases, the noise-to-noise and cluster-to-noise distance-of-distances become more similar to each other. The dimensionality in this example was $D = 20$. The number of neighbors w.r.t which distance-of-distances were computed was 10. The error bar indicates the standard deviation across simulations with different initialization settings.

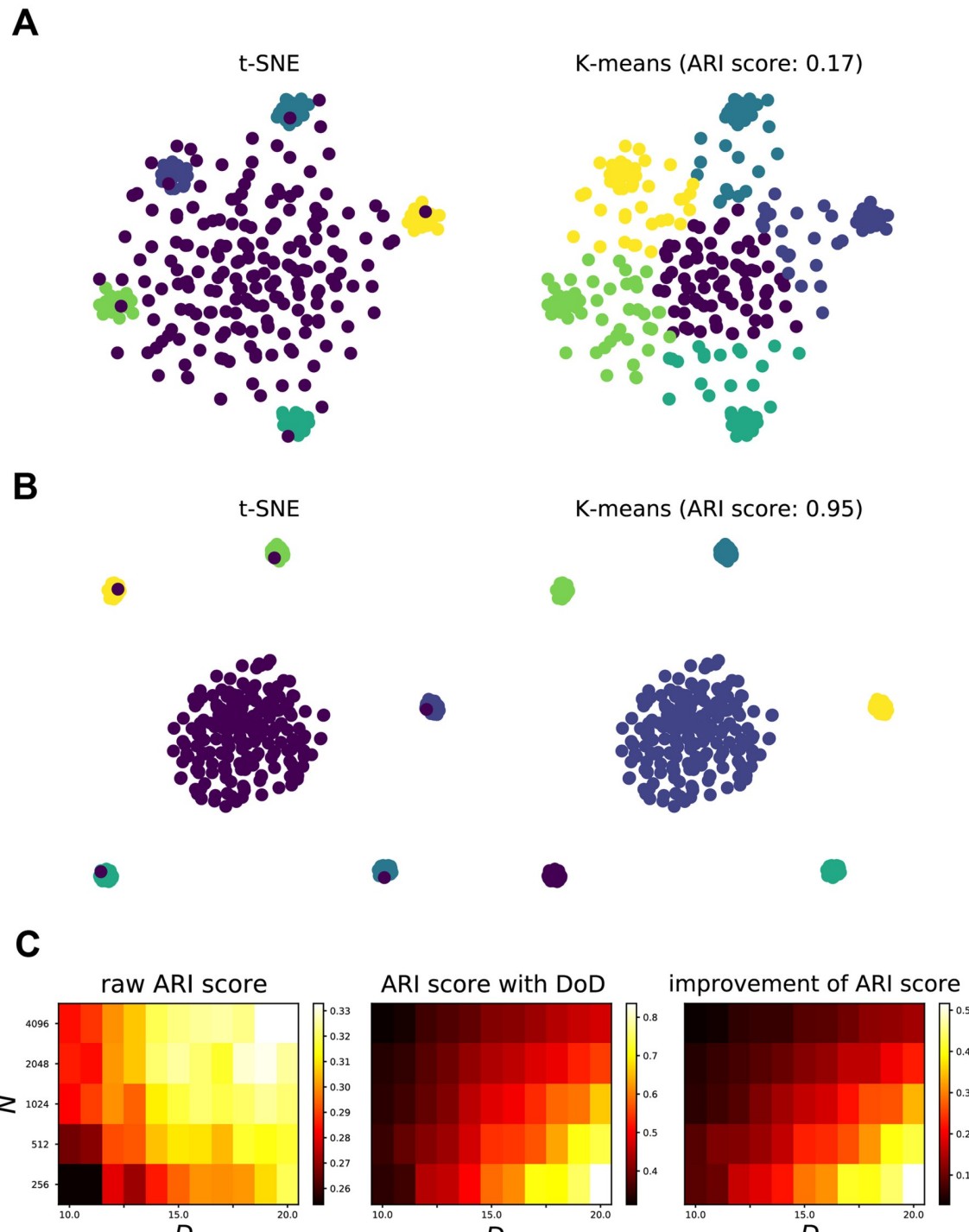

**Fig 4. DoD transformation improves K-means clustering, with a linear dependence on dimensionality and a logarithmic dependence on the number of data points. A**: Example of K-means clustering. We sampled 20 cluster points from five different multivariate Gaussian distributions with dimensionality $D = 20$. Another 500 noise points were sampled uniformly from the same space. *Left*, t-SNE visualization. *Right*, K-means clustering based on t-SNE embeddings. Colors correspond to the original labels. The ARI score of clustering was 0.17. **B**: Distance-of-distance transformation improves the clustering. *Left*, t-SNE visualization of the distance-of-distance matrix. *Right*, K-means clustering. The ARI score of clustering was 0.95. **C**: Effect of distance-of-distance on ARI score, as a function of the number of data points and the dimensionality. Data points were sampled accordingly with different number of noise points and dimensionality $D$. Dimensionality $D$ varied from 10 to 20 linearly. The number of noise points $N$ varied from 256 to 4096 exponentially with a base of 2. *Left*, The ARI score of K-means clustering on the t-SNE embeddings of the original distance

matrix. *Middle*, The ARI score of K-means clustering on the 2D t-SNE embeddings of the distance-of-distance matrix. *Right*, The improvement of ARI score, i.e. difference between the left and middle matrix, which shows the predicted linear dependence on $D$ and logarithmic dependence on $N$.

### Application of DoD transformation to the representation of drifting gratings by mouse visual cortex

We then applied the DoD transformation to high-dimensional empirical data. As a first application, we applied the DoD transformation to the problem of unsupervised detection of spiking sequences in high-dimensional neural data. Previously, techniques have been developed for unsupervised detection of high-dimensional spiking sequence patterns, by using a distance measure between spike trains based on optimal transport (SPOTDist) [14, 17]. By definition, the SPOTDist measure only considers the temporal relationships between spike trains, but is invariant to a scaling of the firing rate [14]. We used this technique to analyze the visual cortical data of Allen Institute Brain Observatory [13]. We wondered whether drifting grating stimuli moving in opposite directions would be represented by different temporal spiking sequences (Fig 5A). The drifting grating stimulus consists of a full-field sinusoidal grating that moves in a direction perpendicular to the orientation of the grating. In the public dataset provided by Allen Institute, the drifting grating stimulus move in 8 different directions (S7 Fig). Furthermore, we also wondered if the neural representations of these stimuli would be similar to the neural vectors of spontaneous activities during the inter-stimulus-interval (Fig 5B). This was motivated by previous studies suggesting a relationship between spontaneous and stimulus-driven or task-evoked neural activity [18–21]. For each trial, we analyzed the responses in the first 100 ms after the stimulus onset and then used the SPOTDist method to compute the pairwise distance between spiking patterns (Fig 5C, Left). Using the SPOTDist distance matrix, the standard t-SNE algorithm revealed a separation of neuronal spiking patterns responding to drifting grating stimulus of different orientations. However, with the standard t-SNE, epochs of spontaneous activities were located in the same region of the low-dimensional embedding as the stimulus-evoked responses (Fig 5D, Left). This visualization seems to suggest that there is relatively high similarity between spontaneous activity and stimulus-driven activity, and that there is some form of replay or preplay of the different stimulus patterns in the inter-stimulus period. Alternatively, the similarity might have been a consequence of the scattering noise problem described above. Consistent with the latter interpretation, we found that the DoD transformation separated the spontaneous activity epochs from the stimulus-evoked epochs. After the DoD transformation, the low-dimensional embedding contained a region for the spontaneous activity that was clearly separated from the stimulus clusters (Fig 5D, Right). This indicates that population vectors during spontaneous activities and activities evoked by drifting gratings are clearly distinguishable, and that the DoD transformation effectively identifies this separation. It also corresponds with previous findings in rodents that the spontaneous activities are living in a space orthogonal to the evoked visual response [22]. In addition, we now observed a clearer separation between the different stimuli. Using KNN to quantify the classification of stimulus orientations, we found that the performance score was improved after the DoD transformation (S8 Fig). Moreover, in the absence of scattering noise, we also showed that the application of DoD transformation did not introduce distortion to the continuous structure in the data (S9 and S10 Figs). However, we observed that the neural response to gratings moving in different directions were not separated both before and after the DoD transformation. The lack of direction separation is due to the fact that the majority of recorded neurons from primary visual cortex have only orientation tuning but not direction tuning

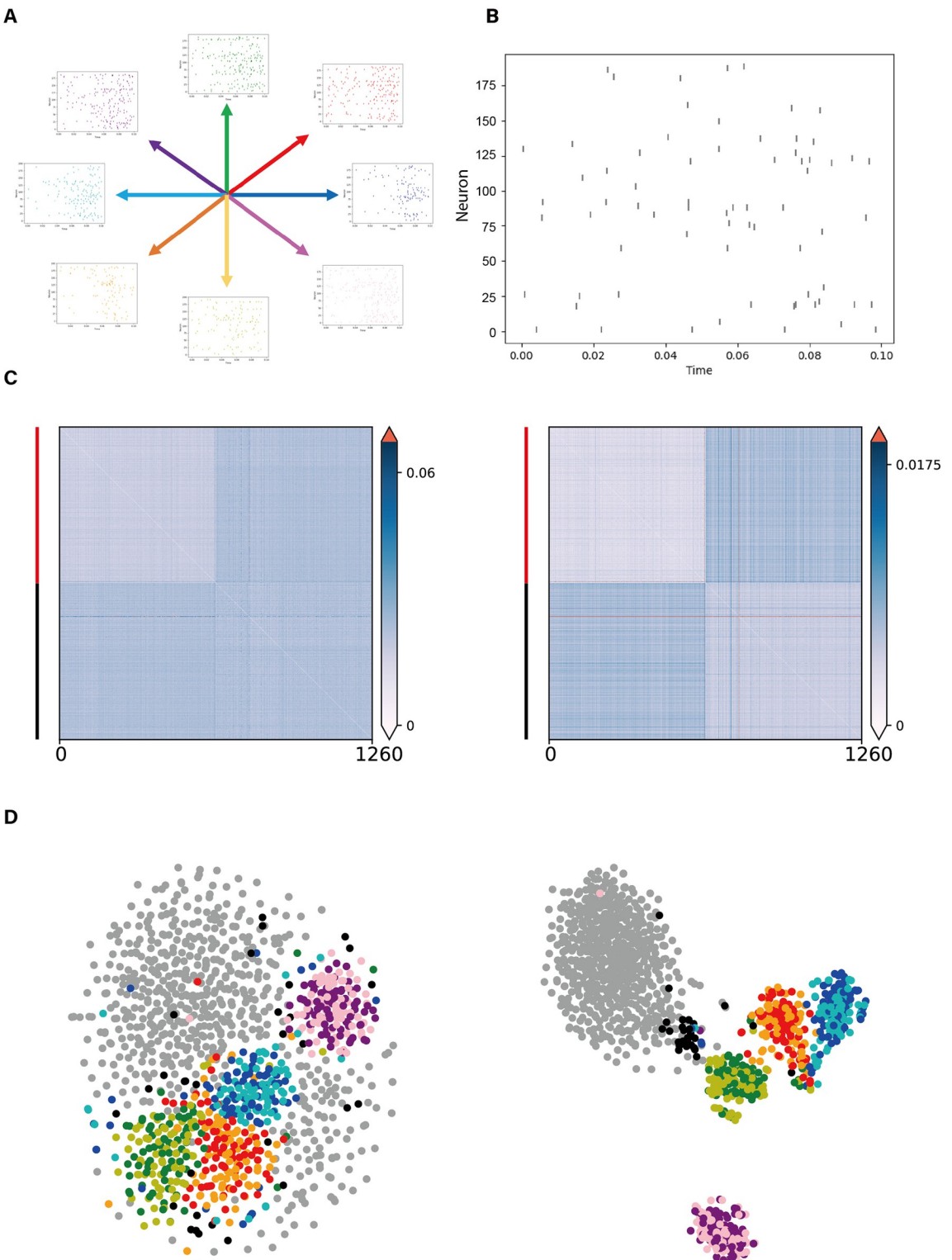

**Fig 5. DoD transformation improves clustering of neural spiking sequences. A**: Example raster plot of spiking pattern to drifting gratings of eight different directions. We analyzed the spiking events in the time window of 100 ms after the stimulus onset. **B**: Example raster plot of spontaneous neural response. We analyzed the spiking events in the time window of 100 ms in intertrial intervals (number of neurons: 191). **C**: Pairwise distances between all spiking patterns. *Left*, Original SPOTDist matrix between spiking patterns. SPOTDist is a distance measure that compares the similarity of the spiking patterns based on optimal transport distance (i.e. the

minimum energy to transform one spiking pattern into another spiking pattern). Total number of stimulus-driven trials was 630. *Right*, Distance matrix after the DoD transformation. **D**: Low-dimensional embeddings of all spiking patterns. *Left*, 2D embeddings of t-SNE on the SPOTDist matrix between spiking patterns. *Right*, 2D embedding of t-SNE on the distance-of-SPOTDist matrix. Drifting grating trials with different directions are labelled in different colors as in **(A)**. Spontaneous activities are colored in grey. Trials with missing labels are colored in black.

(S7(B) Fig). To summarize, we demonstrated that using DoD transformation, we can solve the scattering noise problem that exists in real neural data.

### Application of DoD transformation to the representation of natural images by convolutional neural network

Next, we analyzed the high-dimensional representations of natural images by VGG16, which is a common convolutional neural network used for object recognition [15]. By transformations of input data through multiple convolutional layers, VGG16 represents an image in the fully connected layer as a high-dimensional feature vector of length 4096. A linear classifier can be built upon these feature representations in deeper layers to decode the object identity. We defined our feature vector as the activations of artificial neurons in the fully connected layer of a pre-trained VGG16 network. We then computed the t-SNE embeddings based on these feature vectors of image patches. We first analyzed t-SNE embeddings for image patches that contain 8 out of 1000 different object classes from ImageNet data set. The t-SNE embeddings showed a clear clustering for different object classes (Fig 6A). We then randomly selected 250 image patches from the remaining 992 classes. We only took one image from each distinct class, therefore, it is assumed that these images would scatter in the t-SNE embeddings and mask the clusters, i.e. creating the scattering noise problem. Indeed, we observed that the t-SNE embedding coordinates for these randomly chosen image patches overlapped with the clustered image patches (Fig 6B). Because VGG16 representations live in a high-dimensional space, we predicted that the DoD transformation should lead to a separation of these noise-like scattering image patches from the clustered images. Indeed, the DoD transformation relocated the activation patterns into a separate region of the low-dimensional embedding, while preserving the geometry of object relationships as compared to the original t-SNE (Fig 6C). Using KNN algorithm, we quantified how classification accuracy of the clustering points varied after the DoD transformation. We calculated distance matrices by using either the high-dimensional neural network representations, or the low-dimensional t-SNE embeddings. We found that after the DoD transformation, the cross-validated KNN accuracy increased from 84.3% to 91.2% on high-dimensional neural representations, and it increased from 85.9% to 90.1% on the low-dimensional embeddings. Moreover, applying DoD transformation to data without scattering noise did not distort the clustering patterns (S11 Fig). Furthermore, we also showed that the scattering noise problem could also occur even when the clustering points came from a different image data set (S12 Fig). Therefore, in the case where the clustering data were masked by the scattering noise data, the DoD transformation can better separate them from each other.

### Discussion

We have presented a technique to improve the performance of low-dimensional embedding techniques like t-SNE in the presence of scattering noise points. Such a situation can be common for high-dimensional empirical data where clusters are sparse and a part of the data points represent noise, which may often be the case in biological data. For example, if we were to observe brain activity for several hours, then neural activity may form clear patterns only for

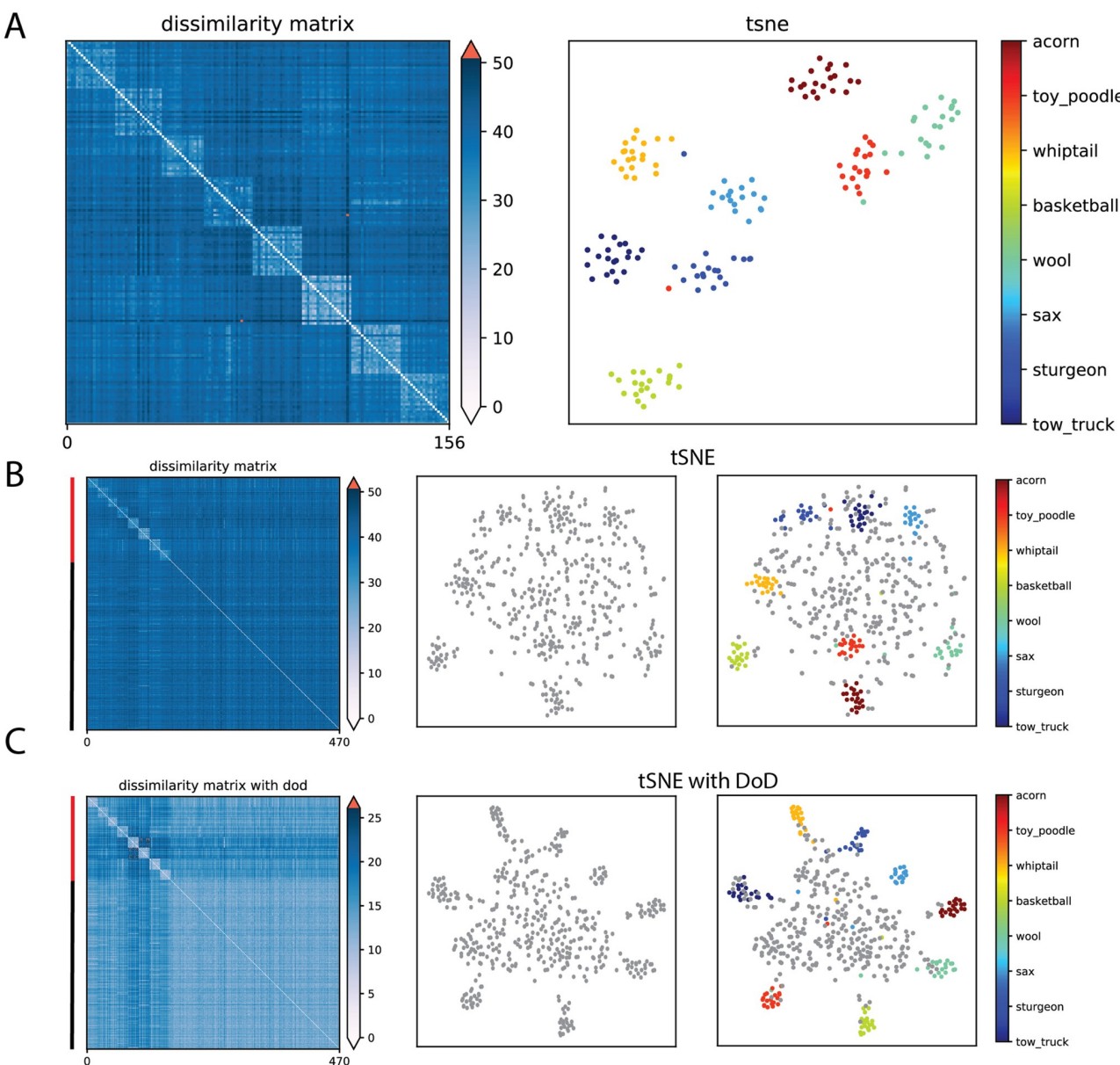

**Fig 6. DoD transformation on natural image patch representations by convolutional neural network units. A**: Low-dimensional manifold for images from 8 different classes. Distances were computed based on the high-dimensional feature vectors in the fully connected layer of the VGG16 network. **B**: Low-dimensional manifold for data included both image from the chosen classes and random images. **C**: DoD transformation separates clustering images from randomly scattering images. *Left*, Dissimilarity matrix. *Middle*, 2D t-SNE embedding without labeling. *Right*, 2D t-SNE embedding with class labeling. 8 classes are labelled in different colors and scattering images are labelled in grey.

a fraction of time, e.g. when neurons are activated by an external stimulus in the receptive field. Importantly, the DoD transformation will yield comparable performance when the data contains only true clusters. Moreover, as we showed, the DoD transformation confers benefits especially when the dimensionality of the data is high. For high-dimensional data, embedding techniques like t-SNE have benefits compared to techniques like PCA, because they can create linear embeddings also for data living on a non-linear manifold, and do not restrict analysis to a few components representing only a fraction of the total variance. Therefore, this technique

can be useful for analyzing the geometry of neural representations because it yields low-dimensional coordinates, which can then be related to other behavioral or stimulus parameters.

A disadvantage of the DoD transformation is its computational cost, because it requires computation of the entire $N \times N$ distance matrix first. Thus, the runtime of the transformation increases exponentially with the number of data points (S13 Fig). By contrast, efficient algorithms exist for computing t-SNE based on neighborhood distances, avoiding an $N \times N$ computational complexity. Another point of consideration is the hyper parameter of the total number of neighbors used to compute the DoD transformation. Although we observed strong improvements with relatively small neighborhood sizes, increasing the neighborhood size beyond the cluster size may lead to distortions (S1 Fig).

In conclusion, we have presented a simple and theoretically motivated transformation of the distance matrix by computing distance-of-distances, which improves clustering of high-dimensional data in the presence of noise points, and have provided several applications to neural networks and biological data where this technique was useful and led to more accurate conclusions.

## Supporting information

**S1 Text. Influence of neighborhood size.**
(PDF)

**S2 Text. Unsupervised noise detection.**
(PDF)

**S3 Text. PCA preprocessing.**
(PDF)

**S4 Text. Fewer noise points than cluster points.**
(PDF)

**S5 Text. Influence of perplexity.**
(PDF)

**S6 Text. Distortion in noise-free situations.**
(PDF)

**S7 Text. Allen Institute Brain Observatory electrophysiological recordings.**
(PDF)

**S8 Text. Improvement of classification.**
(PDF)

**S9 Text. Distortion of real neural data by DoD transformation.**
(PDF)

**S10 Text. Distortion of real convolutional neural network data by DoD transformation.**
(PDF)

**S11 Text. Application of DoD transformation to CNN representation of images from different data sets.**
(PDF)

**S12 Text. Runtime of DoD transformation.**
(PDF)

**S1 Fig. Effect of neighborhood size *K*. A**: 5 clusters, each with 20 points; 200 scattering noise points; dimensionality of 50; Original embedding (left) and DoD transformation with a neighborhood size of 5, 20, 50 and 100. **B**: 50 clusters, each with 20 points; 1000 scattering noise points; dimensionality of 50; Original embedding (left) and DoD transformation with a neighborhood size of 10, 50, 1000 and 2000.
(PNG)

**S2 Fig. Inference of noise points based on neighborhood overlap. A**: Distribution of overlap rate of neighborhood identity before and after the DoD transformation. **B**: Points with smaller overlap rate ($< 65\%$) were identified as scattering noise points (black).
(PNG)

**S3 Fig. PCA is limited in terms of solving scattering noise problem.** 50 clusters, each with 20 points; 1000 scattering noise points; dimensionality of 50; PCA preprocessing uses the first 10 principal components; DoD transformation uses neighborhood size of 10.
(PNG)

**S4 Fig. DoD transformation keeps its performance when there are fewer noise points.** 50 clusters, each with 20 points; 500 noise points; dimensionality of 50; DoD transformation with a neighborhood size of 5. *Left*, original t-SNE visualization. *Right*, t-SNE visualization with DoD transformation.
(PNG)

**S5 Fig. Larger perplexity of t-SNE algorithm does not solve scattering noise problem.** 50 clusters, each with 20 points; 1000 noise points; dimensionality of 50; DoD transformation with a neighborhood size of 5; perplexity values are 5, 50, 100, 500, 1000 from left to right. **A**: t-SNE on original distance matrix. **B**: t-SNE on distance matrix after DoD transformation.
(PNG)

**S6 Fig. DoD transformation does not distort the clustering. A**: 5 clusters, each with 20 points; dimensionality of 50; DoD transformation with neighborhood size ranging from 5 to 30. **B**: 5 clusters, each with 20 points; dimensionality of 50; DoD transformation with neighborhood size ranging from 5 to 30. All clusters were generated from multivariate Gaussian distribution and one of them is with a larger standard deviation (0.5) than the rest (0.2).
(PNG)

**S7 Fig. Neural spiking data from Allen Institute. A**: Illustration of recording session of drifting grating visual stimulus. **B**: Direction tuning curves of recorded units in session 754829445.
(PNG)

**S8 Fig. DoD transformation improves KNN classification of neural spiking sequences.** For both the distances in the high-dimensional space and the distances in the low-dimensional embedding, the cross-validated KNN classification scores are higher after DoD transformation.
(PNG)

**S9 Fig. DoD transformation keeps the ring structure of neural representations of grating stimulus.** Each data point represents the populational firing pattern in a given trial ($n = 3200$). Trials with different stimulus orientations are labelled in different colors.
(PNG)

**S10 Fig. Neural manifolds were better separated by DoD transformation.** Each data point represents the populational spiking pattern in a given trial ($n = 600$). Trials with different stimulus orientations are labelled in different colors.
(PNG)

**S11 Fig. Object manifolds were not distorted by the DoD transformation.** 2D t-SNE embeddings for 8 random ImageNet classes maintain clustering patterns after the DoD transformation.
(PNG)

**S12 Fig. DoD transformation on sketch images represented by convolutional neural network units. A**: Low-dimensional manifold for 634 sketch patches from 5 different classes. Distances were computed based on the high-dimensional vectors in the fully connected layer of the pretrained Alexnet network. **B**: Low-dimensional manifold for data including both sketch patches and 50 random ImageNet patches. **C**: DoD transformation separates ImageNet patches from sketch patches. *Left*, Euclidean distance matrix. *Right*, 2D t-SNE embeddings. Sketch images of different classes are labelled in different colors. ImageNet patches are labelled in grey.
(PNG)

**S13 Fig. Runtime of the DoD transformation.**
(PNG)

## Author Contributions

**Conceptualization:** Jinke Liu, Martin Vinck.

**Data curation:** Jinke Liu.

**Formal analysis:** Jinke Liu.

**Funding acquisition:** Martin Vinck.

**Investigation:** Jinke Liu, Martin Vinck.

**Methodology:** Jinke Liu.

**Project administration:** Martin Vinck.

**Resources:** Martin Vinck.

**Software:** Martin Vinck.

**Supervision:** Martin Vinck.

**Validation:** Jinke Liu.

**Visualization:** Jinke Liu.

**Writing – original draft:** Jinke Liu.

**Writing – review & editing:** Martin Vinck.

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
