## [Decision Letter · Decision Letter 0]

12 Dec 2021

Dear Prof. Dr. Vinck,

Thank you very much for submitting your manuscript "Noise-robust low-dimensional embedding using distance-of-distance transformation" for consideration at PLOS Computational Biology.

As with all papers reviewed by the journal, your manuscript was reviewed by members of the editorial board and by several independent reviewers. In light of the reviews (below this email), we would like to invite the resubmission of a significantly-revised version that takes into account the reviewers' comments.

In particular, reviewer 3 raises major concerns that certain features of tsne, that are traditionally used to address the same problem, may have accidentally been turned off or not implemented. Therefore it is imperative that the authors address this concern both in the manuscript and also ideally through the open release of their code. Both reviewer 2 and reviewer 3 raised major concerns over the clarity of the paper and the transparent benchmarking of the method against t-sne in more real world situations with more testing of the impact of changing different parameters.

We cannot make any decision about publication until we have seen the revised manuscript and your response to the reviewers' comments. Your revised manuscript is also likely to be sent to reviewers for further evaluation.

Sincerely,

Emma Claire Robinson

Associate Editor

PLOS Computational Biology

Thomas Serre

Deputy Editor

PLOS Computational Biology

Reviewer's Responses to Questions

**Comments to the Authors:**

Reviewer #1: The paper by Liu & Vinck suggests a method to transform a NxN pairwise distance matrix such that "noise" points get smaller distances between each other. As a result, when the transformed distance matrix is used for low-dimensional visualization, such as t-SNE, then all noise points get assembled into one "island". This can be convenient as it allows to visually separate noise points. The authors apply their method to spike train recordings and show that it helps distinguish evoked activity and spontaneous activity.

I found the method interesting and the experimental demonstrations convincing, and the paper can be a good fit to PLoS Comp Bio. At the same time, I believe major revision is needed to improve the presentation clarity.

MAJOR ISSUES

* Intuition paragraph (lines 44-50) is very unclear. Sentence in lines 44-46 -- why is that true? Should this be intuitive? It's written as if this sentence should be self-evident, but I think it isn't. Line 47: "to other noise points" -- should this be "to neighbouring noise points"? Is this statement about _all_ noise points or only _neighbouring_ noise points? Line 50: this verbal formulation can be unclear without a formula, I suggest to move the formula and exact definition of your new distance from the Methods here.

* How do the results depend on K? This is never shown. Consider example in Figure 1 (which is very impressive by the way). What happens if you use K smaller than 10? K larger than 10? K equal to the sample size? This needs to be shown. I am actually unsure whether K=n (where n is sample size) will work or will fail. This is very important, and should be explained in the intuition paragraph (see above).

* Are there any downsides of using the transformation? This is never discussed, but should be. In particular, what happens with the noise-free dataset if it is transformed? The authors suggest that nothing much would change, but they need to show direct evidence/quantification. What if the noise-free dataset has some continuous structures? What if one of the clusters has low density (but is still well isolated from the other clusters)? Can you take some real-world noise-free data and show t-SNE embeddings before/after the transformation?

* Continuing previous point: this can be done in Figure 5. If you apply your transformation to the data in panel A, do t-SNE/UMAP become worse or not? You could show it. You could also quantify it by computing kNN classification accuracy, not only in the t-SNE/UMAP embedding, but also directly based on distances (before t-SNE).

* Figure 1 and subsequent simulation Figures (e.g. Figure 2) always have many more noise points than non-noise points. Will the algorithm work as well if noise points are only a small fraction of the total sample size? Please show that.

* line 92: can variance be approximated as well? Using the variance could strengthen the analysis.

* line 188: "clearer separation" -- is this some t-SNE effect or a real effect in the distances matrix? To answer that, you could use kNN classification accuracy, or do before/after t-SNE embeddings of only evoked response points.

* Most of the Methods (sections 4.1-4.2) is textbook description of t-SNE. Frankly, it can be removed, or condensed to a small pargaraph.

* Section 4.3 should be moved to Results, it basically consists of 1 formula. The notation is very confusing! D is used for dimensionality and also to denote distance matrix. k was called K before. Why two notations for the set of nearest neighbors? You don't need to define nearest neighbors, so formulas 18 are unnecessary. \\matcal N is never defined. In equation 19, do you divide by 2k or by the size of the I \\cup J? The sum should be over union of I and J which is not denoted by {I,J}.

* Can the distance matrix after transformation be used for clustering?

* Can your method somehow infer which of the points actually corresponded to noise? Take Figure 1: imagine I don't know that grey points are noise. Can the algorithm identify that?

MINOR ISSUES

* Ref [4] is a strange choice in line 8. This paper does use t-SNE but it's a very minor application there, while scRNA-seq literature has a lot of papers that use t-SNE/UMAP more prominently, to visualize much more complex datasets.

* Introduction is too short, sloppy in places, and does not give literature overview:

a) What is "density-based" in line 20? How is t-SNE density-based?

b) line 22: "maximally preserving pairwise distances" -- sloppy. t-SNE does not aim to preserve distances at all.

c) "crowding problem" -- unfortunate terminology, as the original 2008 t-SNE paper uses the same term "crowding problem" to refer to something else!

d) there are too few references. Are there any related papers at all? That deal with noise in the data? If not in the context of dimensionality reduction, then maybe clustering? If you cannot find any relevant research whatsoever, at least say so.

* line 58: "data poits" -> "noise data points"?

* Figure 2: add column titles indicating dimensionality.

* line 82: K was used for number of neighbors, now it's number of clusters -- confusing

* line 111: "while preserving distance between the clusters and the noise points" -- that's sloppy formulation, these distances are not exactly preserved.

* Figure 3: what are error bars?

* Figure 3: unclear what is the cluster variance used for GMM simulation -- it should be some realistic data, not infinitely dense.

* Figure 6, panel B: y-axis ticks are cropped

* Figure 6, panel C: could you label "signal" and "noise" parts of the distance matrix?

* Figure 6, panel D: what are the black points?

* line 204: please mention some memory and runtime requirements. How long does it take for what sample size.

* line 206: "further work is needed..." -- is this at all feasible? Sounds rather unfeasible to me.

* line 210: "distortions" -- this needs to be shown, see my comment above about choice of K

* Section 4.4: what t-SNE implementation was used? Give version. And UMAP?

* line 271: "different initialization" -- what initialization?

* line 273: "small variance" -- this is insufficient level of detail. Please describe all your experiments EXACTLY. Make it clear what refers to what experiment (which figure) exactly.

Reviewer #2: I have attached my review as a word document.

Reviewer #3: In this paper, Liu and Vinck describe an interesting strategy for obtaining more informative 2D embeddings of high-dimensional data. The strategy is simple: convert the distance matrix typically used by embedding methods into a distance of distances matrix. This appears to be very useful on two real-world datasets, and the simulated data also drives the point home. I have two major concerns for this manuscript.

Major points:

1) There is an automatic rescaling of distances in t-SNE/UMAP based on the number of neighbors each point has, via the perplexity parameter. This should take care of the relative density problem that this paper deals with. I would like to make sure that this automatic rescaling was enabled, and that the authors always determined a separate sigma_j for each point. However, the code was not shared, so I cannot check this directly. Furthermore, there are many Euclidian distances presented to us throughout the manuscript, which makes me think the distances were not re-estimated. Furthermore, the theoretical analyses indeed do not consider the rescaling of distances implicit in t-SNE/UMAP.

If sigma_i was indeed estimated directly for each point as it should be, then I would like to know why the authors think that was not enough. If it was estimated, then I think a lot of the visualizations should change to show the transformed similarity matrices instead of the Euclidian distances, since the transformed similarity matrices are the actual data that t-SNE uses. The theoretical analysis itself probably has to be redone to introduce the sigma_j's in section 2.1. Confusingly, sigma here takes a different meaning, so that variable name has to change too.

I am not sure how you are running t-SNE, but if you're passing in a similarity matrix directly to an open implementation of t-SNE, then you would have needed to do the point-by-point rescaling by yourself. There is way too little information in the methods about what you did, so please add all those details there as well. There are some other important parameters in t-SNE and UMAP, you need to specify those in your Methods section.

2) The problem with tSNE and UMAP is that there are many tricks for making them work better, and even when you do all the tricks right, there are many aspects of the embedding that seem arbitrary. Some computational researchers reject this outright, claiming that tSNE and UMAP are bad representations of the data and are just there to make "pretty pictures". I wouldn't go as far, I think there's some inherent use for these visualizations, and there are some consistent ways to get good embeddings (see The Art of tSNE). If the authors can compare their tricks to these more standard and consistent tSNE methods, then I think the paper can be a useful contribution to our bag-of-tricks for tSNE and UMAP.

In particular, the authors have to show that the crowding problem persists when the tSNE is run in this more standard way: reduce dimensions by PCA, initialize tSNE with PCA, use a perplexity of approx N/100 where N is the number of points. They should also show that the problem persists with different tSNE parameters, such as higher or lower perplexity. Also, UMAP has its own set of parameters that needs to be explored, and all the parameter settings have to be specified in the methods.

Minor points:

1) heatmaps in all images should be scaled in a more reasonable way (except 4c which is fine) . Right now they just all look red. Perhaps scale between 1% and 99% saturation?

2) the analysis in 2.5 is interesting, but you should mention that the orthogonality of stimulus and spontaneous activity was shown in Stringer et al, Science 2019 for rodents. The activity patterns literally live in different linear subspaces.

3) Also with respect to 2.5, a comment should be made about the lack of separation of patterns to opposite directions of motion. We know there is direction selectivity in V1, but embedding algorithms have to throw some information away, and the direction selectivity seems to indeed disappear.

4) You should add the information about dimensions and number of points to the figure so it's clear without carefully reading the legend. Or at least separate the first two columns from the last two columns so the reader doesn't think it's a progression of 4 settings for one parameter like I originally thought.

5) Line 228: this is just not true. Choosing the perplexity is one of the most important decision when running t-SNE, as well as a few other things like initialization with PCA and how many PCs to keep from the data (see The Art of TSNE).

**Have the authors made all data and (if applicable) computational code underlying the findings in their manuscript fully available?**

Reviewer #1: None

Reviewer #2: **No: **They have stated the code and data are available on github, but the link has not been provided.

Reviewer #3: **No: **They haven't shared the code yet.

PLOS authors have the option to publish the peer review history of their article (what does this mean?). If published, this will include your full peer review and any attached files.

Reviewer #1: **Yes: **Dmitry Kobak

Reviewer #2: **Yes: **Alex Diaz-Papkovich

Reviewer #3: No
---

## [Decision Letter · Decision Letter 1]

28 Sep 2022

Dear Prof. Dr. Vinck,

Thank you very much for submitting your manuscript "Improved visualization of high-dimensional data using the distance-of-distance transformation" for consideration at PLOS Computational Biology. As with all papers reviewed by the journal, your manuscript was reviewed by members of the editorial board and by several independent reviewers. The reviewers appreciated the attention to an important topic. Based on the reviews, we are likely to accept this manuscript for publication, providing that you modify the manuscript according to the review recommendations.

Sincerely,

Emma Claire Robinson

Academic Editor

PLOS Computational Biology

Thomas Serre

Section Editor

PLOS Computational Biology

[LINK]

Reviewer's Responses to Questions

**Comments to the Authors:**

Reviewer #1: The revision is a MASSIVE improvement, with all raised points adequately addressed. I wish I saw such a thorough and comprehensive revision more often! I only have minor comments left.

MINOR COMMENTS

Fig S1A -- to illustrate the point better, you could remove K=10 and K=15 from the top row (K=5 and K=20 are nearly identical anyway) and rather add K=50 and K=100, which would show how the performance degrades for larger K.

Fig S1A -- replace "original" -> "original embedding"? The word "original" suggests to me that this panel shows the original 2D data which is of course not the case as the generated data are higher-dimensional

Fig S6 -- "K" instead of "number of neighbors"? That's how it was in Fig S1

Fig S6 -- corr coefficients can be rounded e.g. to two or three digits, currently the precision looks odd

Fig S7 -- here I am confused: are the simulated data 2D or higher-dimensional? If the data are higher-dimensional, then I don't understand why the "original" t-SNE shows smaller density for the lower-left cluster. t-SNE adjusts the kernel width to reach a given perplexity and typically does not preserve the cluster density.

"introduces very limited distortions and preserve"  "... preserves"

Fig S12 -- red dots in the matrix look really weird and confusing. If it's simpler all values above the top colorbar limit, then I would suggest to color them dark blue (same shade as in the top end of the colorbar).

"In situations where there are less noise than the number of clusters" -- cumbersome formulation. "... where the number of noise points is smaller than the number of non-noise points"?

Fig S2 -- what was the K here? Same as for DoD computation?

"(e.g. distances among correlation matrices, or optimal transport distances over spiking patterns [13].)" -- remove the period

line 69: unclear why you use L1 distance here and not L2. Same about line 72. Would it work with L2 distance in either the first, or the second, or both places? Could be nice to show that empirically e.g. for the situation in Figure 1

Fig 4C -- it is confusing to me that N grows from top to bottom and not from bottom to top...

line 222 -- please state here what the dimensionality is (4096?)

Reviewer #2: The authors have addressed the points from my previous review adequately. The article is well-presented and interesting, however it will need to be proofread for spelling and grammar errors.

-Inconsistencies in writing "neighbour" versus "neighbor", "DoD" versus "Distance-of-Distance", etc

-Author summary: typo with "hihg-dimensional". I believe it should also be "low-dimensional" instead of "low dimensional" to be consistent

-The paragraph at 173 is unclear and needs to be rewritten. It has several grammar and spelling errors: "less noise points than..." instead of "fewer noise points than...", "our method are" instead of "our method is", "one cluster has low density than" instead of "one cluster has lower density than"

-381: should be "requires computation of the..."

-384: should be "an NxN..."

-393: I believe the authors mean to convey that their method provided improved or more accurate conclusions rather than simply "qualitatively different" conclusions (at least that's how I see it!)

Reviewer #3: The authors have addressed all my comments in a satisfactory manner. Thank you.

**Have the authors made all data and (if applicable) computational code underlying the findings in their manuscript fully available?**

Reviewer #1: **No: **Please make the entire code available on Github

Reviewer #2: Yes

Reviewer #3: Yes

PLOS authors have the option to publish the peer review history of their article (what does this mean?). If published, this will include your full peer review and any attached files.

Reviewer #1: **Yes: **Dmitry Kobak

Reviewer #2: **Yes: **Alex Diaz-Papkovich

Reviewer #3: No

Figure Files:

Data Requirements:

Reproducibility:

References:

---

## [Decision Letter · Decision Letter 2]

10 Nov 2022

Dear Prof. Dr. Vinck,

Thank you very much for submitting your manuscript "Improved visualization of high-dimensional data using the distance-of-distance transformation" for consideration at PLOS Computational Biology. As with all papers reviewed by the journal, your manuscript was reviewed by members of the editorial board and by several independent reviewers. The reviewers appreciated the attention to an important topic. Based on the reviews, we are likely to accept this manuscript for publication, providing that you modify the manuscript according to the review recommendations.

Sincerely,

Emma Claire Robinson

Academic Editor

PLOS Computational Biology

Thomas Serre

Section Editor

PLOS Computational Biology

Reviewer's Responses to Questions

**Comments to the Authors:**

Reviewer #1: Thanks again to the authors for addressing my comments, and congratulations on very nice work. I recommend acceptance. I only noticed one small issue now:

* in Formula 1, I don't understand why the fraction before the sum is 1/2K, as the size of the I \\cup J union may be smaller than 2K. Should 2K in that formula be replaced by |I \\cup J|?

Reviewer #2: My comments have been adequately addressed.

**Have the authors made all data and (if applicable) computational code underlying the findings in their manuscript fully available?**

Reviewer #1: Yes

Reviewer #2: Yes

PLOS authors have the option to publish the peer review history of their article (what does this mean?). If published, this will include your full peer review and any attached files.

Reviewer #1: **Yes: **Dmitry Kobak

Reviewer #2: **Yes: **Alex Diaz-Papkovich

Figure Files:

Data Requirements:

Reproducibility:

References:

---

## [Decision Letter · Decision Letter 3]

28 Nov 2022

Dear Prof. Dr. Vinck,

We are pleased to inform you that your manuscript 'Improved visualization of high-dimensional data using the distance-of-distance transformation' has been provisionally accepted for publication in PLOS Computational Biology.

Best regards,

Emma Claire Robinson

Academic Editor

PLOS Computational Biology

Thomas Serre

Section Editor

PLOS Computational Biology

Reviewer's Responses to Questions

**Comments to the Authors:**

Reviewer #1: Thanks! I recommend acceptance.

**Have the authors made all data and (if applicable) computational code underlying the findings in their manuscript fully available?**

Reviewer #1: None

PLOS authors have the option to publish the peer review history of their article (what does this mean?). If published, this will include your full peer review and any attached files.

Reviewer #1: **Yes: **Dmitry Kobak

---

## [Editor Report · Acceptance letter]

14 Dec 2022

PCOMPBIOL-D-21-01871R3 

Improved visualization of high-dimensional data using the distance-of-distance transformation

Dear Dr Liu,

I am pleased to inform you that your manuscript has been formally accepted for publication in PLOS Computational Biology. Your manuscript is now with our production department and you will be notified of the publication date in due course.

With kind regards,

Anita Estes
